

# Kinematic characteristics of barefoot sprinting in habitually shod children

Jun Mizushima[1], Keitaro Seki[1,2,3], Justin W.L. Keogh[4,5,6], Kei Maeda[1], Atsushi Shibata[1], Hiroyuki Koyama[7] and Keigo Ohyama-Byun[8]

[1] Graduate School of Comprehensive Human Sciences, University of Tsukuba, Ibaraki, Japan
[2] Neuromuscular Research Center, Faculty of Sport and Health Sciences, University of Jyväskylä, Jyväskylä, Finland
[3] Department of Physical Education, College of Humanities and Sciences, Nihon University, Tokyo, Japan
[4] Faculty of Health Sciences and Medicine, Bond University, Gold Coast, Australia
[5] Sports Performance Research Centre New Zealand, Auckland University of Technology, Auckland, New Zealand
[6] Cluster for Health Improvement, Faculty of Science, Health, Education and Engineering, University of the Sunshine Coast, Sunshine Coast, Australia
[7] Faculty of Education, Kyoto University of Education, Kyoto, Japan
[8] Faculty of Health and Sport Sciences, University of Tsukuba, Ibaraki, Japan

Corresponding author
Jun Mizushima, d50h60@gmail.com

## ABSTRACT

**Background**. Anecdotally, a wide variety of benefits of barefoot running have been advocated by numerous individuals. The influence of the alterations in the properties of the shoe on the running movement has been demonstrated in adults at submaximal jogging speeds. However, the biomechanical differences between shod and barefoot running in children at sprinting speeds and the potential developmental implications of these differences are still less examined. The purpose was to determine the potential differences in habitually shod children's sprint kinematics between shod and barefoot conditions.

**Methods**. Ninety-four children (51 boys and 43 girls; 6–12 years-old; height, $135.0 \pm 0.12$ m; body mass, $29.0 \pm 6.9$ kg) performed 30 m maximal sprints from standing position for each of two conditions (shod and barefoot). To analyze sprint kinematics within sagittal plane sprint kinematics, a high-speed camera (300 fps) was set perpendicular to the runway. In addition, sagittal foot landing and take-off images were recorded for multiple angles by using five high-speed cameras (300 fps). Spatio-temporal variables, the kinematics of the right leg (support leg) and the left leg (recovery leg), and foot strike patterns: rear-foot strike (RFS), mid-foot strike (MFS), and fore-foot strike (FFS) were investigated. The paired $t$-test was used to test difference between shod and barefoot condition.

**Results**. Barefoot sprinting in habitually shod children was mainly characterized by significantly lower sprint speed, higher step frequency, shorter step length and stance time. In shod running, 82% of children showed RFS, whereas it decreased to 29% in barefoot condition. The touch down state and the subsequent joint movements of both support and recovery legs during stance phase were significantly altered when running in condition with barefoot.

**Discussion**. The acute effects of barefoot sprinting was demonstrated by significantly slower sprinting speeds that appear to reflect changes in a variety of spatiotemporal parameters as well as lower limb kinematics. It is currently unknown whether such differences would be observed in children who typically run in bare feet and what

developmental benefits and risks may emerge from increasing the proportion of barefoot running and sprinting in children. Future research should therefore investigate potential benefits that barefoot sprinting may have on the development of key physical fitness such as nerve conduction velocity, muscular speed, power, and sprinting technique and on ways to minimize the risk of any acute or chronic injuries associated with this activity.

## INTRODUCTION

Running is the one of the most fundamental motor skills, which typically emerges around 2 years after the birth (*Gallahue, 1982*). Children are commonly believed to achieve motor skills in a specific developmental sequence throughout the course of the central nervous system (CNS) maturation and anatomical development of bones, muscles, and other soft tissues (*Forssberg, 1999*). Upon reaching the age of 6–7 years, the running movement of children exhibits many of the same spatio-temporal characteristics as adults (*Miyamaru, 2001*).

It has been known that running movement is influenced by the alterations in the properties of the shoe (*De Wit, De Clercq & Aerts, 2000*; *Lieberman et al., 2010*). Habitually shod adults runners, for example, have a higher step frequency, while having shorter step lengths and contact times when running barefoot (*De Wit, De Clercq & Aerts, 2000*). In addition, habitually shod adults and adolescents runners are more likely to utilize a rear-foot strike when wearing shoes, with this shifting toward mid-foot or fore-foot when no wearing shoes (*Lieberman et al., 2010*).

It has been proposed that one of the reasons for these differences between barefoot and shod running is that shoes limit proprioception (*Lieberman, 2012*). Sensory feedback from the plantar surface of the foot has evolved as an adaptation for sensing characteristics of the ground including hardness, roughness and unevenness (*Lieberman, 2012*). Planter proprioception is responsible for transmission of information about the contact surface characteristics to the CNS, thereby regulating key aspects of the emerging gait patterns (*Fiolkowski et al., 2005*; *Nurse & Nigg, 2000*). However, proprioception feedback may be acutely reduced due to wearing shoes, and this may affect whole running movement through alteration of sensory feedback. If this acute reduction in proprioception during one bout of shod running was to be repeated consistently throughout a child's development, there may be the potential that the child may not develop adequate foot proprioception and that this may limit their ongoing development and physical maturation.

Another possible factor influencing shod running is the mass of the shoe and how it influences the recovery leg's inertial characteristics particularly in the swing phase. From a biomechanical perspective, the moment of inertia around the hip joint is affected by the mass of each segment of the lower limb and the distance of this mass from the hip joint. The foot, the most distal limb segment, is swung around the largest radius of gyration in recovery leg as pendular action. Thus, a shoe's mass may have considerable effect on the

movement of swinging the recovery leg due to inertial resistance. The shoe-related increase in the leg's moment of inertia would increase the hip flexor torque requirements during the swing phase. If these increased hip flexor torques are unable to be maintained for the required race distance, this would result in a reduced hip angular velocity which may alter critical aspects of running technique including key aspects of foot strike. Specifically, these changes in foot strike may include alterations in horizontal distance from the whole body's centre of gravity (CG) to the center of mass of support leg's foot and perhaps differences in the point of contact between the foot and ground. As a shoe's mass can be a high proportion of foot mass for children who have an immature musculoskeletal system, its influence could be more profound during childhood than adulthood.

Although the differences between barefoot and shod running have been well examined, almost all of this research has involved adults at submaximal jogging speeds (*De Wit, De Clercq & Aerts, 2000*; *Lieberman et al., 2010*; *Muñoz-Jimenez et al., 2015*). There are substantially less research examined on this topic in children, especially at sprinting speeds (*Latorre-Román, Balboa & García-Pinillos, 2017*; *Latorre-Román et al., 2018*), and kinematic analysis at sprinting speeds has never been researched in detail. Sprinting is required in many sport activities (*Mero, Komi & Gregor, 1992*; *Rossi et al., 2017*) and the regular performance of sprinting activities may contribute to the development of key sensorimotor and physical fitness qualities such as reaction time, muscular speed and power. Therefore, the aim of this study was to determine the potential acute differences in habitually shod children's sprint kinematics when sprinting barefoot compared to shod.

## METHODS

### Participants

Ninety-four habitually shod children (51 boys and 43 girls; age, 6–12 years-old; height, 1.35 ± 0.12 m; body mass, 29.0 ± 7.0 kg) participated in this study (Table 1). Height and body mass was quite similar to standard level of Japanese children with the same age (*MEXT, 2017*). Written informed consent was obtained from each participant and their parents. The Ethics Committee of the Faculty of Health and Sports Sciences at the University of Tsukuba approved this study (IRB ID: 29-143). The inclusion criteria were as follows: (i) all of them were free from physical and/or intellectual disabilities and free of injuries at the time of the experiment, (ii) they were habitually shod children with no experience in barefoot running, (iii) all participants were from a community sports club and had been practicing basic skills of running, jumping and throwing through 18 days per year.

### Data collection and processing

After warming up, the participants performed maximal-effort 30 m sprints from standing position in two conditions (shod and barefoot) on the rubber surface of runway. Running conditions were randomized for each child in order to compare the shod and barefoot condition.

Figure 1 shows the experimental setup. Two-dimensional coordinates on sagittal plane of anatomical landmarks were obtained using a high-speed camera (EX-F1, CASIO, Tokyo, Japan). The frame rate and shutter speed of the camera were set at 300 fps and 1/1000
**Table 1 Physical characteristics of participants.**

| Age | n (Boy/Girl) | Height (m) | Weight (kg) |
|---|---|---|---|
| 6 | 7 (3/4) | 1.17 ± 0.4 | 20.0 ± 1.6 |
| 7 | 13 (6/7) | 1.22 ± 0.2 | 22.3 ± 0.9 |
| 8 | 14 (10/4) | 1.29 ± 0.7 | 25.9 ± 3.2 |
| 9 | 15 (6/9) | 1.35 ± 0.6 | 28.6 ± 4.3 |
| 10 | 25 (13/12) | 1.39 ± 0.5 | 30.8 ± 4.2 |
| 11 | 13 (10/3) | 1.47 ± 0.9 | 36.6 ± 7.1 |
| 12 | 7 (3/4) | 1.52 ± 0.7 | 39.6 ± 6.5 |

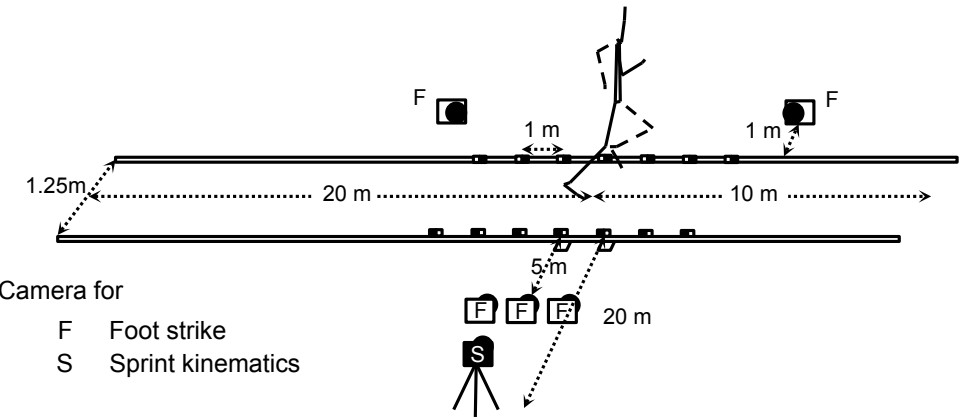

Camera for

F    Foot strike
S    Sprint kinematics

**Figure 1 Experimental setup.**

s, respectively. The camera was placed 20 m apart from the right side of the 20 m point of the runway such that the optical axis was perpendicular to the plane of movement. Reference markers were set on both sides of the runway at one-meter intervals (17–23 m) and recorded for reconstructing two-dimensional coordinates of the body from images. In addition, foot strike patterns were evaluated using five high-speed cameras (EX-F1, CASIO, Tokyo, Japan) placed on rightward, forward diagonal and backward diagonal direction of 20 m point of runway. All images of foot landing were captured on the hard disk drive of a MacBook Pro (Apple Inc., Cuptertino, CA, USA).

Twenty-three anatomical landmarks (3rd metacarpal head of hands, styloid processes of ulnas, elbows, shoulders, 1st toes, head of 5th metatarsal bones, heels, ankles, knees, greater trochanters, head, ears, and suprasternal) during two steps (from the left foot touchdown to the next touchdown of the same foot) were digitized at 150 Hz with a motion analysis system (Frame-DIAS VI, DKH Inc., Tokyo, Japan). The coordinates of anatomical landmarks were smoothed with a Butterworth low-pass digital filter at optimal cut-off frequencies (5.0–10.0 Hz) determined by residual analysis (*Wells & Winter, 1980*). Fourteen segment models comprising hands, forearms, upper arms, feet, shanks, thighs, head and trunk was obtained by using smoothed coordinates. We used the estimations of

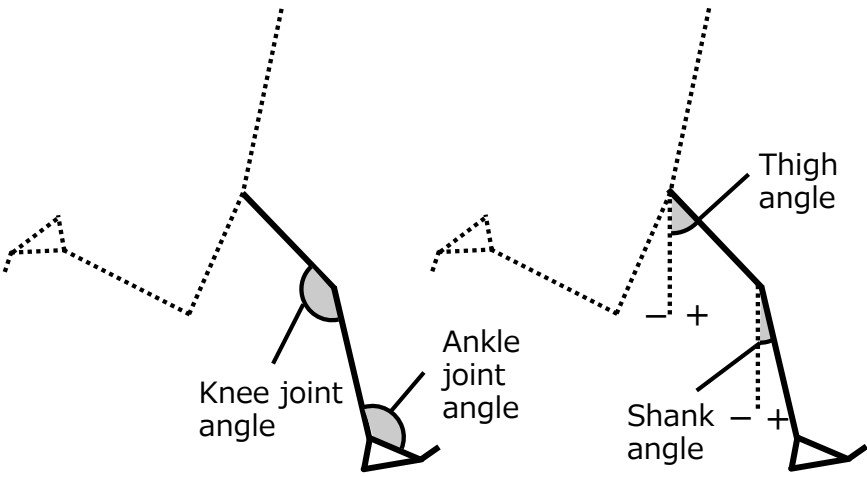

**Figure 2  Definitions of calculated angles.**

*Yokoi, Shibukawa & Ae (1986)* to measure the position of the center of mass and moment of inertia of each segment.

## Analysis

The frames that a foot touches and leaves the ground were defined as touchdown and takeoff, respectively. Stance phase was the period when either the left or right foot was in contact with the ground, and flight phase was the period when neither foot was in contact with the ground.

Step frequency is the reciprocal of the time required for one step. Step length is the horizontal traveling distance of the whole body's CG for one step. Sprint speed is the product of step frequency and step length. Step frequency, step length, sprint speed, stance time, and flight time were averaged for two consecutive steps. Each time was measured and calculated visually from the filming rate (150 Hz).

The kinematics of the right leg (support leg) and the left leg (recovery leg) during the stance phase of right foot (the angles and the angular velocities of the thigh, shank, knee joint, and ankle joint) were calculated. Thigh and shank angles were defined as the angle between each segment and the vertical line. Positive value was assigned when the distal end of the segment was positioned in front of the proximal end (Fig. 2). The time series data for all participants were normalized by the time of stance phase in shod condition.

We visually classified foot strikes into three patterns (Fig. 3): rear-foot strike (RFS), mid-foot strike (MFS), and fore-foot strike (FFS) by using the definition of *Lieberman (2012)*. RFS was defined as a landing in which the heel lands before the ball of the foot (a heel-toe run). FFS was defined as a landing in which the ball of the foot lands before the heel (a toe-heel-toe run). MFS was defined as a simultaneous landing of the heel and ball of the foot. These identifications were conducted with QuickTime Player 7 Pro for Mac (Apple Inc., Cuptertino, CA, USA).
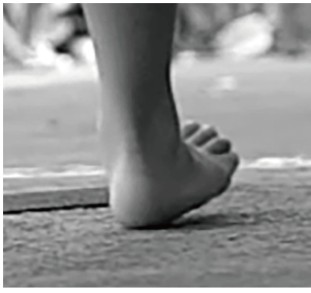 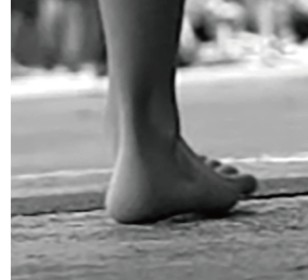 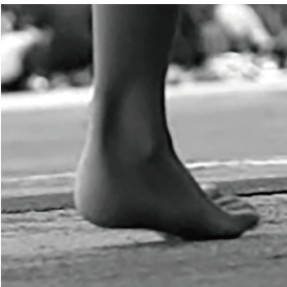

(a) Rear-foot strike (RFS) (b) mid-foot strike (MFS) (c) fore-foot strike (FFS)

**Figure 3 Definitions of foot strikes.** (A) Rear-foot strike (RFS); (B) mid-foot strike (MFS); (C) fore-foot strike (FFS).

## Statistical analysis

Results were presented as mean ± standard deviation (SD) as the assumptions of normality were met according to the results of a Shapiro-Wilk test, paired $t$-test were used to examine the potential differences in sprint kinematics between shod and barefoot condition with the effect sizes (Cohen's d) and 95% confidence intervals (CI). McNemar's test was used to examine the difference in foot strike patterns between shod and barefoot condition with the odds ratio (OR) and 95% confidence intervals (CI). To analyze the effect of age and sex on sprint speed and foot strike pattern, one-way ANOVA, independent sample $t$-test and chi-square analyses were used. The statistical significance level was set at 5%. All statistical procedures were conducted using the statistical package SPSS for Mac version 25 (SPSS Inc., Armonk, NY, USA).

## RESULTS

### Spatio-temporal variables

Table 2 demonstrates spatio-temporal variables for the entire sample. One-way ANOVA indicated a significant effect of age on sprinting speed ($p < 0.001$), with the faster runners being older; while result of independent sample $t$-test indicated no significant effect of sex on sprinting speed ($p = 0.859$). On the other hand, chi-square analyses indicated no significant effect of age or sex on foot strike pattern ($p = 0.054–0.195$). Sprint speed was significantly lower in barefoot than in the shod condition ($d = 0.19$, 95% CI [$-0.10–0.48$], $p = 0.004$). Step frequency was significantly higher ($d = -0.38$, 95% CI [$-0.66–0.09$], $p < 0.001$) and step length significantly shorter ($d = 0.42$, 95% CI [$0.13–0.70$], $p < 0.001$) in barefoot than in the shod condition. Stance time was significantly shorter in the barefoot condition ($d = 0.55$, 95% CI [$0.26–0.84$], $p < 0.001$), while no significant difference was found in flight time ($d = -0.08$, 95% CI [$-0.37–0.20$], $p = 0.397$).

### Foot strike pattern and sagittal plane kinematics of support leg

Table 3 demonstrates foot strike pattern and the lower limbs kinematics at touchdown. Eighty-two percent of children showed RFS in shod running, whereas this decreased to 29% in the barefoot condition (OR = 8.46, CI [$1.84–38.90$], $p < 0.001$). Angle of the ankle joint

**Table 2  Spatiotemporal variables.**

|  | Shod | Barefoot | *t* value |
|---|---|---|---|
| Sprint speed (m/s) | 5.60 ± 0.62 | 5.48 ± 0.70[*] | 2.977 |
| Step frequency (Hz) | 3.98 ± 0.34 | 4.11 ± 0.37[**] | −4.681 |
| Step length (m) | 1.47 ± 0.17 | 1.34 ± 0.20[**] | 6.958 |
| Stance time (s) | 0.148 ± 0.018 | 0.139 ± 0.017[**] | 6.671 |
| Flight time (s) | 0.105 ± 0.019 | 0.107 ± 0.019 | −0.850 |

Notes.
[**]$p < 0.01$.
[*]$p < 0.05$.

**Table 3  Foot strike pattern and the lower limb kinematics of the support leg at touchdown.**

| Condition | Foot strike (%) | | | Joint angle (deg) | | Joint angular velocity (deg/s) | | CG to foot distance (m) |
|---|---|---|---|---|---|---|---|---|
|  | RFS | MFS | FFS | Ankle | Knee | Ankle | knee |  |
| Shod | 82 | 15 | 3 | 101.2 ± 6.1 | 143.2 ± 7.2 | 159.2 ± 195.4 | 27.3 ± 159.4 | 0.24 ± 0.05 |
| Barefoot | 29 | 43 | 28 | 103.3 ± 6.1[**] | 144.4 ± 6.8 | −38.0 ± 165.7[**] | −104.4 ± 131.9[**] | 0.23 ± 0.05[*] |
| *t* value |  |  |  | −3.109 | −1.382 | 7.783 | 6.721 | 2.291 |

Notes.
[**]$p < 0.01$.
[*]$p < 0.05$.

at touchdown was significantly greater in barefoot than in the shod condition ($d = −0.34$, 95% CI −0.63 to −0.06, $p = 0.002$). There was no significant difference at the knee joint angles at touchdown ($d = −0.16$, 95% CI [−0.44–0.13], $p = 0.397$). Angular velocities of the ankle ($d = 1.09$, 95% CI [0.78–1.39], $p < 0.001$) and knee joints ($d = 0.90$, 95% CI [0.59–1.19], $p < 0.001$) were significantly greater in shod than in the barefoot condition and these showed negative values (for dorsiflexion and flexion) in barefoot condition, whereas positive values (for plantarflexion and extension) in the shod condition. The horizontal distance from CG to the center of mass of support leg's foot was significantly shorter in barefoot than in the shod condition ($d = 0.29$, 95% CI [0.00–0.58], $p = 0.024$).

## Sagittal plane kinematics of recovery leg

Table 4 demonstrates angle and angular velocity of the thigh during the stance phase. The thigh angle at takeoff was significantly less in barefoot than in the shod condition ($d = 0.40$, 95% CI [0.11–0.69], $p < 0.001$), while no significant difference was found at touchdown ($d = −0.21$, 95% CI [−0.49–0.08], $p = 0.138$). The range of motion of the thigh angle during the stance phase was significantly smaller in barefoot than in the shod condition ($d = 0.47$, 95% CI [0.18–0.75], $p < 0.001$). The thigh angular velocity at touchdown was significantly higher in barefoot than in the shod condition ($d = −0.46$, 95% CI [−0.75−−0.17], $p < 0.001$). Figure 4 shows the time-series variations of the knee joint angle. The knee joint angle was greater after the mid-stance until the takeoff in barefoot than in the shod condition.

**Table 4  Thigh angles and thigh angular velocity of recovery leg in stance phase.**

| Condition | Thigh angle (deg) | | | Thigh angular velocity (deg/s) |
|---|---|---|---|---|
| | Touchdown | Takeoff | Range of Motion | Touchdown |
| Shod | −15.8 ± 8.4 | 60.6 ± 5.7 | 76.4 ± 8.1 | 428.5 ± 98.0 |
| Barefoot | −14.1 ± 8.4 | 58.1 ± 6.7[**] | 72.2 ± 9.8[**] | 474.8 ± 104.0[**] |
| *t* value | −1.496 | 3.685 | 3.474 | −3.468 |

Notes.
[**]$p < 0.01$.

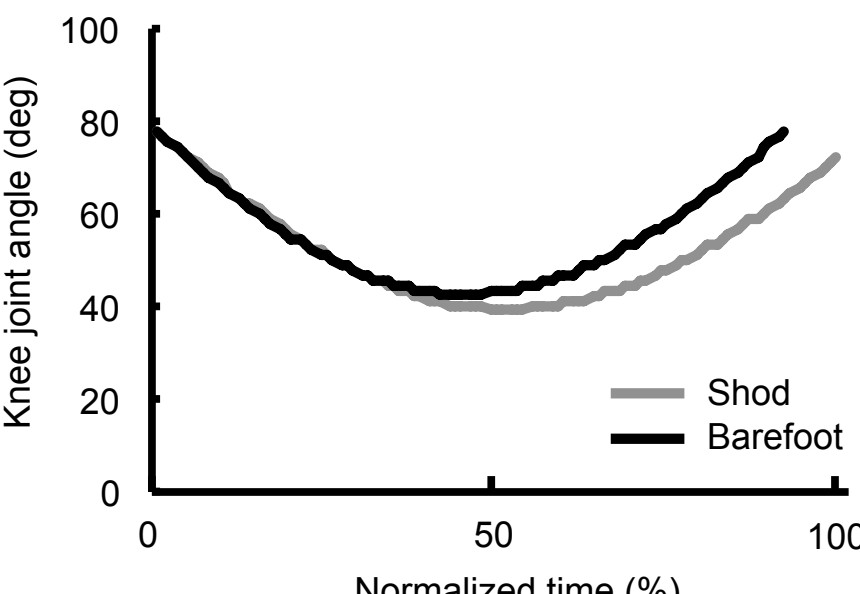

**Figure 4  Time-series variations of the knee joint angle of recovery leg during the contact phase.** Time was normalized by the stance time of shod condition.

## DISCUSSION

The purpose of this study was to quantify the potential acute differences in habitually shod children's sprint speed and kinematics when sprinting barefoot compared to shod. In general, sprint kinematics were changed when running barefoot in a similar way to the literature for adults barefoot running at lower relative velocities (*De Wit, De Clercq & Aerts, 2000*; *Lieberman et al., 2010*).

   The major findings of the current study were that a significantly lower sprint speed, higher step frequency and shorter step length were observed in barefoot condition. These results are consistent with those previously found by *De Wit, De Clercq & Aerts (2000)* in nine adults at similar relatively absolute speeds (3.5, 4.5 and 5.5 m s$^{-1}$) to the current study (∼5.5–5.6 m s$^{-1}$). Results of the current study also indicated that stance time was significantly shorter in barefoot than in the shod condition, although there was no significant difference in flight time. Step frequency is composed of stance time and flight time (*Hay, 1993*). Thus, the result of current study indicated that increased step frequency

in barefoot condition was a result of a significant reduction in stance time. As observed in earlier research (*Hasegawa, Yamauchi & Kraemer, 2007*), shorter contact time may necessitate a more mid- or fore-foot strike pattern.

Such a premise was supported by the results of the current study, whereby 69% of the children with RFS in shod condition shifted to FFS or MFS in barefoot condition. The reduction of RFS prevalence reinforces the results of previous study (*Latorre-Román, Balboa & García-Pinillos, 2017*) in children, which observed a RFS prevalence of 83.95% during shod running compared to 62.65% when running barefoot. This change in foot strike pattern was consistent with the ankle joint angle, which was significantly more plantarflexed at touchdown in barefoot condition. These changes of the initial conditions at touchdown may reflect the requirement to reduce the localized pressure on the heel based on previous findings for adult runners (*De Wit, De Clercq & Aerts, 2000*). When children sprint barefoot and land with a RFS, the longitudinal arch of the foot does not lengthen until after the ball of foot lands (*Perl, Daoud & Lieberman, 2012*). In this position, the magnitude of loading directly impacts the heel pad and it is suddenly deformed to a physiological maximum (*De Clercq, Aerts & Kunnen, 1994*). On the other hand, the localized pressure on the heel can be avoided by adopting FFS or MFS landings, through which initial ground contact covers a larger planter area (*De Wit, De Clercq & Aerts, 2000*). This will allow many lower limb joints including the longitudinal arch of the foot to absorb the impact force in a FFS and MFS. While it is acknowledged that the study is cross-sectional in nature and involved children who typically sprinted with shoes, an implication of this result is that if these children were to perform some additional barefoot sprinting they may acquire greater proprioceptive feedback and utilize a greater number of lower limb joints and anatomical structures to receive ground contact forces.

Furthermore, some studies of barefoot running investigate the relationship between the initial kinematic condition at touchdown and the vertical impact force peak. *Lieberman et al. (2010)* argued that a RFS landing causes large impact transients, whereas a FFS landing typically generate ground reaction forces that lack a distinct transient in adults during shod and barefoot running at 3.5 m s$^{-1}$. In the present study, the ankle joint was dorsiflexing and the knee joint was flexing at touchdown in barefoot condition, while plantarflexing and extending in the shod condition. This motion used during barefoot sprinting may be an adaptation to reduce the vertical impact force peak. However, no significant difference was observed in flight time between barefoot and shod condition. Even though sprinting speed was significantly reduced during barefoot sprint, it could be assumed that children obtained similarly large ground reaction forces as a result of their shorter stance time in barefoot condition as in shod condition.

The assumption that barefoot condition induced efficient transmission and/or receipt of force toward/from the ground in children sprinting could be discussed from a stretch-shortening cycle exercise and muscular adaptation perspectives. *Ahn et al. (2014)* reported that FFS runners activate plantarflexors earlier than RFS runners and it increases the capacity of the passive structures to store elastic energy at the beginning of stance. When children sprint barefoot and land in a FFS or a MFS with only an ankle dorsiflexion during initial stance phase, the plantarflexor muscles would stretch passively and store elastic

energy. This enhances the force output of plantarflexors at takeoff. These changes may also contribute to the preferential utilization of stretch-shortening cycle and development of hypertrophy and force power velocity characteristics of the plantarflexor muscles, because eccentric loads allow the generation of more muscle force, power and velocity than isolated concentric actions (*Lieberman, 2012*). This suggests that even though an acute decrement in performance occurs when sprinting barefoot, the utilization of some barefoot sprinting may help children develop the ability to engage the plantarflexor muscles during the rapid stretch-shortening cycles that may result in an increase in the ground reaction forces during the stance phase of sprinting (*Nagahara et al., 2018*; *Rossi et al., 2017*). However, such increased forces acting through the plantarflexors may also increase the risk of injuries found in runners such as Achilles tendon strains or gastrocnemius tears. This suggests that the incorporation of barefoot sprinting into the overall physical activity program for children needs to be gradually progressed in a systematic fashion if they are to alter their running technique and to minimize their risk of injury is not to be increased (*Murphy, Curry & Matzkin, 2013*; *Tam, Tucker & Astephen Wilson, 2016*).

Another issue to consider is the effect of mass of shoes. The average mass of the foot of the participants is approximately 470 g estimated by usage of inertial characteristics determined by *Yokoi, Shibukawa & Ae (1986)*. Assuming that the mass of general running shoes for children is 150–250 g per shoe, the mass of shoes constituted a high proportion of overall foot-shoe mass. Such a change in the inertial characteristics may produce relatively large increases in the lower limbs moment of inertia, which could negatively affect the angular velocity of the recovery leg in the swing phase. Such a view was consistent with the results whereby, the forward angular velocity of the thigh was significantly higher in barefoot than in the shod condition at touchdown in this study. Consequently the knee extension in barefoot condition is significantly larger than in the shod condition. These changes can be explained by the effect of the increased mass of the shoes and its subsequent increase on the moment of inertia around the proximal joints. *Seki et al. (2016)* examined mechanical role of the recovery leg during sprinting for elementary school children, and they suggested that increasing forward swing velocity of the recovery leg contributes large mechanical energy flow between recovery and support legs. Increased transferring mechanical energy between both legs would be efficient to exert force on the ground during shorter stance time. Thus, in barefoot condition, greater forward swing velocity of the recovery leg may therefore lead to improved sprint technique and performance in the long-term.

Finally, there are also a number of significant differences in the kinematics of both support and recovery leg as a whole between the two sprinting conditions. The horizontal distance from CG to the center of mass of support leg's foot was significantly shorter in barefoot. This would suggest a reduction in the range of motion swept by the support leg and recovery leg, which may have also contributed to the shorter stance time, higher step frequency and shorter step length when sprinting barefoot.

From a practical point of view, physical education teachers or sports coaches may wish to include some proportion of barefoot sprinting in children's physical activity and sporting programs for the development of key physical fitness such as muscular speed, power, neuromuscular coordination and sprinting technique. In addition, this activity needs only

taking the shoes off and does not involve any additional effort or expensive/cumbersome exercise equipment. However, there are still some aspects to consider about the potential risk of injuries. First, safe environment for barefoot sprinting must be prepared because injurious debris or temperature extremes may cause acute injury to the foot surface (*Murphy, Curry & Matzkin, 2013*). Second, the proportion of barefoot sprinting should be increased slowly. The alteration of foot strike pattern from RFS to FFS or MFS can increase metatarsal and Achilles tendon stress and therefore increase the risk of running injury (*Murphy, Curry & Matzkin, 2013*).

While we recruited a larger sample of participants than most previous studies, there were still a number of limitations to the study's design. The first is that in order to gain a sample of 94 habitually shod children, males and females between the ages of 6-12 years were recruited. As there are a variety of anatomical, physiological and biomechanical differences between males and females and as substantial maturation can occur between the ages of 6 to 12 years, it is possible that some of our results may have been affected by these interactions of sex and age. Moreover, as the experiment was conducted on rubber surface of specific runway, it may not be directly applicable to sprinting on other surfaces such as grass or the sand found at beaches where children are more likely to run barefoot.

## CONCLUSION

In conclusion, this study demonstrated a significant reduction in sprinting speed when barefoot that reflected a number of significant differences in sprint spatiotemporal parameters and lower body kinematic parameters. While it is unknown whether such differences would be observed in children who typically run barefoot, the developmental benefits and risks that may emerge from increasing the proportion of barefoot running and sprinting in children require additional research. Future research should therefore investigate potential benefits that barefoot sprinting may have on the development of sensorimotor function and key physical fitness parameters, while also looking to minimize the risk of any acute or chronic injuries associated with this activity.

## ACKNOWLEDGEMENTS

We would like to thank the participants and the staff of community sports club (Kyo$^2$ club) from Kyoto University of Education who helped to collect the data.

### Funding

The authors received no funding for this work.

### Competing Interests

Justin W.L. Keogh is an Academic Editor for PeerJ.

## Author Contributions

- Jun Mizushima conceived and designed the experiments, performed the experiments, analyzed the data, prepared figures and/or tables.
- Keitaro Seki analyzed the data, contributed reagents/materials/analysis tools, prepared figures and/or tables, authored or reviewed drafts of the paper.
- Justin W.L. Keogh authored or reviewed drafts of the paper, approved the final draft.
- Kei Maeda analyzed the data, prepared figures and/or tables, authored or reviewed drafts of the paper.
- Atsushi Shibata and Hiroyuki Koyama performed the experiments, contributed reagents/materials/analysis tools.
- Keigo Ohyama-Byun conceived and designed the experiments, authored or reviewed drafts of the paper, approved the final draft.

## Human Ethics

The following information was supplied relating to ethical approvals (i.e., approving body and any reference numbers):

This study was approved by the Ethics Committee of the Faculty of Health and Sports Sciences at the University of Tsukuba (29-143).

## Data Availability

The raw data have been supplied as Data S1.

## Supplemental Information

Supplemental information for this article can be found online at http://dx.doi.org/10.7717/peerj.5188#supplemental-information.

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
