# Peer review of "Kinematic characteristics of barefoot sprinting in habitually shod children"

_PeerJ, doi:10.7717/peerj.5188_

## Round 0.1 · original submission · Major Revisions

Dear authors:

Your manuscript was evaluated by expert reviewers. Please address reviewer´s concerns in detail.

Also, take this chance to check your manuscript conforms to the various PeerJ criteria, which I reiterate below:

-improve the writing style and clarity of your paper.
-make sure to adapt your manuscript to PeerJ policies.
-use clear and unambiguous text that conforms to professional standards.
-include sufficient introduction and background to demonstrate how the work fits into the broader field of knowledge, with relevant prior literature appropriately referenced.
-use the format of ‘standard sections’ from PeerJ Instructions for Authors (a significant modification from PeerJ suggested structure should be made only if this modification significantly improve clarity or conform to a discipline-specific custom).
-please double check that your research question is clearly defined, is relevant and meaningful, identifying the knowledge gap being investigated and how the study contributes to filling that gap.
-please double check to provide the Methods section with sufficient information to be reproducible by another investigator.
-review that your data be robust and statistically sound.
-assure that your conclusions be appropriately stated, connected to the original question investigated, and limited to those supported by the results.
-speculation (especially in the discussion of your results) is welcomed, but should be identified as such.

Reviewer 1 ·

Basic reporting

The article should include enough introduction and background to demonstrate how the work fits into the broader field of knowledge. The references are very old, you should include important references such as the following:

Latorre-Román, P. Á., Párraga-Montilla, J. A., Guardia-Monteagudo, I., & García-Pinillos, F. (2018). Foot strike pattern in preschool children during running: sex and shod–unshod differences. European journal of sport science, 1-8.

Latorre-Román, P. Á.L., Balboa, F. R., & García-Pinillos, F. (2017). Foot strike pattern in children during shod-unshod running. Gait & posture, 58, 220-222.

Muñoz-Jimenez, M., Latorre-Román, P. A., Soto-Hermoso, V. M., & García-Pinillos, F. (2015). Influence of shod/unshod condition and running speed on foot-strike patterns, inversion/eversion, and vertical foot rotation in endurance runners. Journal of sports sciences, 33(19), 2035-2042.

Paragraph 78-86, how the mass of the shoe influences on the kinematic or FSP parameters?

The limitations and practical applications should be included, for example:
Limitaciones:

Shod/unshod condition must be randomized
The age, sex and antropometric influences should be analyzed
The rubber surface is another limitations

Discussion

The purpose showed in discussion section must be the same expressed in the introduction section. Be consistent with the terms.

Previous studies of Latorre et al., should be taken into account.

Practical aplications:

Why the children should run barefoot at high speed?

Experimental design

Partcipants:

Other inclusion criteria should be included: barefoot experience, level of physical activity, physical and / or intellectual disabilities, etc.

Data collection and processing

How the FPS were analyzed ?, by observational method.

How do you know that the unshod condition could influence on the shod condition and not the other way around? Therefore, shod / unshod condition must be randomized,



Statistical analysis

To analyse the foot strike patterns (FSP) differences between the shod/unshod condition, McNemar’s test must be used

Validity of the findings

No coments

Additional comments

Thank you for the opportunity to review this manuscript. I have listed some specific comments below for your consideration. Please regard them as constructive recommendations. The paper is well written and is methodologically correct.

·

Basic reporting

1a) The use of English is appropriate and the writing style is easy to follow and understand
1b) The literature is scarce. Only 18 references are used, one of them is data of the Japanese Ministry and almost half of the references are from the 80's or 90's. Very relevant research, including very recent from Japanese colleagues who analyzed sprinting mechanics in Japanese children using a 50-m long force platform system is not reference, as well as the works by important authors on this topic such as Morin et al. Both the introduction and discussion should be significantly updated to include more recent and relevant literature.
1c) Article is well structured and the raw data is properly shared
1d) The purpose of the study is clear (i.e., to analyze the potential differences between running conditions in children), and the results are in line with the purpose of the study.

Experimental design

2a) The purpose is relevant and original
2b) Research question is well defined, and it is justified how it adds to the current body of literature (i.e., there are known differences between running barefoot or with shoes in adults, but not in children)
2c) The study follows all ethical standards and the investigation have been performed in a rigorous way
2d) Methods are clear enough so the study could be replicated by another group of researchers

Validity of the findings

3a) No comment
3b) Data is fully available in a spreadsheet. However, statistical analysis are quite weak as they are only based on the NHS and no effect sizes or CI are reported
3c) The authors fail to link the main result of the manuscript (i.e., sprint performance in barefoot condition is impaired) to their conclusion, as they don't critically discuss how this might affect training strategies or how coaches should use this information. They rather discuss how running barefoot could be used to develop proprioception and prevent injuries; variables that were not measured at all in the manuscript, nor was the purpose of the investigation
3d) Claims that running barefoot could potentiate proprioception and prevent injuries are not adequate and are not linked to the purpose of this investigation, since no single variable related to injuries was tested (authors even required participants to not be/have been injured, so potential links to injury prevention in a retrospective way is not possible)

Additional comments

Dear authors, thanks for your interesting manuscript. It is a well written study that covers an unstudied topic (i.e., comparing barefoot vs shod running in Japanese children), but it has some flaws that should be addressed/improved in order to get it accepted. Please follow these recommendations to improve your manuscript. Summarizing, your main problems, that requires major changes, are:

1. References are very scarce. You need much more relevant and RECENT studies, almost half of the studies have 15-20 years or more
2. Statistical analyses are also scarce. There are several limitations of the NHS, and only reporting p-values is not currently considered a good practice. You need to add to your current results, at least, the effect size and its 95% CI.
3. You need to improve your discussion to be better linked to your research question (i.e., the differences between running barefoot or with shoes), because you claim that several times that running barefoot could improve proprioception and prevent injuries, but the only thing that you really observed, and is not properly discussed, is the impairment of sprint performance in the barefoot condition. IMHO, this is the most important result, and the one which has more relevant practical applications for coaches.

BASIC REPORTING
1a) The use of English is appropriate and the writing style is easy to follow and understand
1b) The literature is scarce. Only 18 references are used, one of them is data of the Japanese Ministry and almost half of the references are from the 80's or 90's. Very relevant research, including very recent from Japanese colleagues who analyzed sprinting mechanics in Japanese children using a 50-m long force platform system is not reference, as well as the works by important authors on this topic such as Morin et al. Both the introduction and discussion should be significantly updated to include more recent and relevant literature.
1c) Article is well structured and the raw data is properly shared
1d) The purpose of the study is clear (i.e., to analyze the potential differences between running conditions in children), and the results are in line with the purpose of the study.

EXPERIMENTAL DESIGN
2a) The purpose is relevant and original
2b) Research question is well defined, and it is justified how it adds to the current body of literature (i.e., there are known differences between running barefoot or with shoes in adults, but not in children)
2c) The study follows all ethical standards and the investigation have been performed in a rigorous way
2d) Methods are clear enough so the study could be replicated by another group of researchers

VALIDITY OF THE FINDINGS
3a) No comment
3b) Data is fully available in a spreadsheet. However, statistical analysis are quite weak as they are only based on the NHS and no effect sizes or CI are reported
3c) The authors fail to link the main result of the manuscript (i.e., sprint performance in barefoot condition is impaired) to their conclusion, as they don't critically discuss how this might affect training strategies or how coaches should use this information. They rather discuss how running barefoot could be used to develop proprioception and prevent injuries; variables that were not measured at all in the manuscript, nor was the purpose of the investigation
3d) Claims that running barefoot could potentiate proprioception and prevent injuries are not adequate and are not linked to the purpose of this investigation, since no single variable related to injuries was tested (authors even required participants to not be/have been injured, so potential links to injury prevention in a retrospective way is not possible)

---

## Round 0.2 · accepted · Accept

I am pleased to inform you of the official acceptance of your manuscript for publication in PeerJ.

·

Basic reporting

Ok

Experimental design

Ok

Validity of the findings

Ok

Additional comments

Issues solved